# Improvement in tropospheric moisture retrievals from VIIRS through the use of infrared absorption bands constructed from VIIRS and CrIS data fusion

E. Eva Borbas[1], Elisabeth Weisz[1], Chris Moeller[1], W. Paul Menzel[1], Bryan A. Baum[2]

[1]Cooperative Institute for Meteorological Satellite Studies, University of Wisconsin-Madison, Madison, Wisconsin, USA

[2]Science and Technology Corporation, Madison, Wisconsin, USA

*Correspondence to*: E. Eva Borbas (eva.borbas@ssec.wisc.edu)

**Abstract.** An operational data product available for both the Suomi-NPP and NOAA-20 platforms provides high spatial resolution infrared (IR) absorption band radiances for VIIRS based on a VIIRS+CrIS data fusion method. This study investigates the use of these IR radiances, centered at 4.5, 6.7, 7.3, 9.7, 13.3, 13.6, 13.9, and 14.2 µm, to construct atmospheric moisture products (e.g., total precipitable water and upper tropospheric humidity) and to evaluate their accuracy. Total precipitable water (TPW) and upper tropospheric humidity (UTH) retrieved from hyperspectral sounder CrIS measurements are provided at the associated VIIRS sensor's high spatial resolution (750m) and are compared subsequently to collocated operational Aqua MODIS and Suomi-NPP VIIRS moisture products. This study suggests that the use of VIIRS IR absorption band radiances will provide continuity with Aqua MODIS moisture products.

## 1. Introduction

Retrieval of atmospheric water vapor properties from the Visible Infrared Imaging Radiometer Suite (VIIRS) satellite sensor on the Suomi National Polar-orbiting Partnership (S-NPP) and the National Oceanic and Atmospheric Administration (NOAA-20) platforms is challenging due to the absence of infrared (IR) water vapor absorption bands. Fortunately, measurements in the missing spectral region are available on the Crosstrack Infrared Sounder (CrIS), a hyperspectral IR sensor also on the same platforms. Spectral measurements in these IR absorption bands can be constructed for VIIRS through fusion of the imager and sounder data. Weisz et al. (2017) demonstrated a fusion method to construct IR water vapor and carbon dioxide absorption band radiances for VIIRS at 750m spatial resolution. With the addition of the missing spectral bands to VIIRS on Suomi-NPP, this study evaluates Total column Precipitable Water vapor (TPW) and Upper Tropospheric Humidity (UTH) in clear skies through comparison to the MODerate resolution Imaging Spectroradiometer (MODIS) MYD07 (Borbas et al. 2016) and MYD08 (Platnick et al., 2013) Collection 6.1 and Version 1.0 VIIRS (Borbas et al., 2019a-d) atmospheric products. The MYD07 is a Level-2 swath product that provides temperature and water vapor profiles at 5-km spatial resolution, while the MYD08 provides water vapor on an 8-day global grid at 1˚ x 1˚ resolution. Through comparison to the MYD07/MYD08 products, we will demonstrate that the VIIRS water vapor product shows better agreement when these constructed band radiances are included, with the major improvement being found in the tropics.

While VIIRS has a wide scanning swath, high horizontal resolution, a nearly constant pixel size across the scan, and a day/night band (DNB), it's spectral complement lacks thermal infrared (IR) absorption bands necessary to accurately retrieve tropospheric moisture content as well as cloud properties that rely on those spectral measurements. In particular for moisture retrievals, VIIRS does not take measurements in the broad 6.7-µm water vapor band that are measured by the MODIS (Seemann et al., 2003). Fortunately, the missing IR spectral bands can be gleaned from measurements on the companion hyperspectral CrIS sensor on the same platform.

Here we denote the instantaneous field of regard as field-of-view (FOV) for the sounder and pixel for the imager exclusively to minimize confusion between the two sensors. To achieve TPW and UTH at imager pixel resolution, this study employs the innovative data fusion approach of Weisz et al. (2017) that constructs MODIS-like water vapor and $CO_2$ sensitive radiances directly at the imager resolution through use of co-located VIIRS and CrIS radiances. In this study, the data fusion method provides MODIS-like IR absorption band radiances at the VIIRS M-band spatial resolution (750m). The VIIRS+CrIS fusion radiances are available for the entire record of both S-NPP and NOAA-20 platforms (Baum et al., 2019a).

The availability of these IR-band radiances for VIIRS at 750m pixel resolution makes it possible to retrieve a cloud mask and moisture properties using algorithms developed and tested using the full MODIS spectral band suite (Borbas et al., 2011). The goal of this study is to determine the impact of supplementing VIIRS with imager-resolution VIIRS+CrIS fusion bands on retrieving TPW and UTH and establish the feasibility of extending the MODIS TPW and UTH into the future with those derived from VIIRS+CrIS fusion. The benefits associated with the continuation of such a high spatial resolution product include, for example, the observation of high spatial scale weather phenomena (weather forecasting), the urban heat islands (Hu and Brunsell, 2015), and in determining atmospheric correction for high spatial resolution remote sensing products such as the MODIS land surface temperature products (Proud et al, 2010, Hulley et al, 2017, Wen, 2010).

This paper is organized as follows: Section 2 discusses data and fusion method: Section 3 summarizes the moisture retrieval method and presents results, and a summary of the findings is provided in Section 4.

## 2.    Data and Methodology

The VIIRS sensor is a 22-band scanning radiometer that is currently flying on the NASA Suomi NPP and the NOAA-20 platforms. VIIRS has 16 bands scanning a 3000 km swath at 750m resolution (medium resolution, or M), 5 bands at 375m resolution (imaging, or I), and a day/night band. For this investigation, the focus is on using the bands at M resolution.  The data used in this study include the standard Level–1B VIIRS data for both the S-NPP and NOAA-20 platforms made available by the Atmosphere Science Investigator-led Processing System (A-SIPS) located at the University of Wisconsin–Madison Space Science Engineering Center (SSEC).

The Cross-track Infrared Sounder (CrIS) is a Fourier transform spectrometer with 1305 spectral channels in normal spectral resolution (NSR) and 2211 channels in full spectral resolution (FSR) over 3 wavelength ranges: LWIR (9.14- to 15.38-µm); MWIR (5.71- to 8.26-µm); and SWIR (3.92- to 4.64-µm). CrIS scans a 2200 km swath width (± 50 degrees), with 30 Earth-scene views. Each view consists of 9 FOVs from a 3x3 array that have a nadir spatial resolution of approximately 14 km.

The fusion method requires an accurate co-location between the high spatial resolution imager data (with pixels at 750m) and the lower-spatial-resolution sounder data (with FOVs at about 14 km). The fusion method described in Weisz et al. (2017) consists of two steps for each imager pixel: (a) Search nearby neighbors to find the five FOVs that best match the split window (i.e., 11 and 12-µm) imager pixel radiances averaged over the FOV to an individual pixel's split window measurements – this is accomplished using a *k-d* (or multi-dimensional) tree search algorithm (Bentley, 1975) on both high spatial (M-band 11 and 12-µm data) and low spatial (M-band 11 and 12-µm data averaged over the CrIS FOV) resolution imager radiances. (b) Convolve the high spectral sounder radiances (at low spatial resolution) to the desired IR broadband; then average the convolved sounder radiances associated with the selected five nearest neighbors to construct the desired spectral band for each imager pixel. Spectral radiance convolution refers to the process of converting high spectral resolution (narrowband) to broadband radiance measurements by applying a spectral response function (SRF) of a given broadband. Here, SRFs associated with the spectral bands of the MODIS sensor on the NASA Earth Observation system (EOS) Aqua platform are applied to CrIS measurements. The VIIRS+CrIS fusion IR absorption band radiances are available for the entire records of S-NPP and NOAA-20 at the Level-1 and Atmosphere Archive and Distribution System (LAADS) Distributed Active Archive Center (DAAC) at NASA Goddard Space Flight Center (Baum et al., 2019a, b).

VIIRS+CrIS fusion radiances alongside observed radiances for MODIS bands 25 (4.5 µm), 27 (6.7 µm) and 35 (13.9 µm), repeated from Weisz et al. (2017), are shown in Fig. 1. The fusion results for band 27 show more inaccuracies, because $H_2O$-sensitive spectral bands sense different tropospheric regions than split-window spectral bands. Also, small-scale and narrow spatial features in moisture (e.g., dry slots and cloud edges), which are not captured by the sounder due to its large spatial resolution, are more problematic for the fusion process. Furthermore, the results at the edge of the imager granule (i.e., outside the sounder swath) should be used with caution since they do not account for "limb darkening" and hence tend to be less accurate. Results shown in Fig. 1 for the VIIRS+CrIS fusion radiances compared well in a qualitative sense with the observed MODIS radiances, even in the more challenging water vapor band.

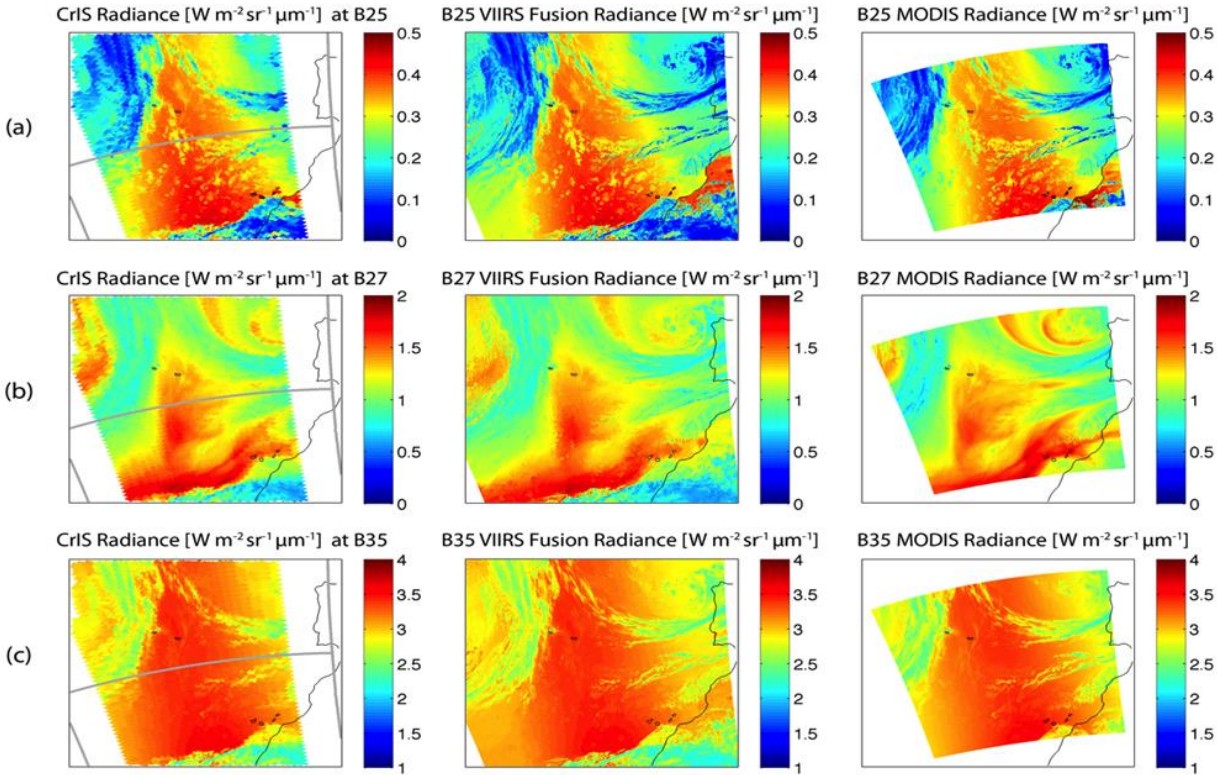

Figure 1. CrIS sounder radiance (left), newly constructed fusion radiance (middle), and the observed MODIS radiance differences (right) for MODIS bands 25 (4.5 µm), 27 (6.7 µm) and 35 (13.9 µm) in panels a, b and c, respectively, for one granule at 1436 UTC on April 17, 2015. This is shown as Figure 8 in Weisz et al. (2017).

To assess the viability of the moisture products to provide continuity with similar products from MODIS, we perform a comparison with co-located measurements (i.e., matchups) with Aqua MODIS. For this study, the co-location process requires the VIIRS 750m pixel to be fully contained within the MODIS 1-km pixel; the scene must be high confidence clear (as identified by the MODIS cloud mask MYD35); and the scan angles for the matching pair must be less than 50˚ so that it is within the swath of the CrIS sensor. Figure 2 shows the results of tens of thousands of instances of collocated MODIS and VIIRS+CrIS fusion radiances that are converted to brightness temperatures (BTs) in two water vapor and four carbon dioxide bands for the month of April 2018. It can be seen that the mean clear-sky brightness temperature differences (BTDs) between VIIRS+CrIS fusion and original MODIS data are less than 0.5 K for MODIS $H_2O$ bands 27 to 28 (6.7 and 7.3 µm) and MODIS $CO_2$ bands 33 to 36 (13.3, 13.6, 13.9, and 14.2 µm) for 11-µm BTs ranging from 200 to 280 K. Root mean square scatter (not shown) about these mean values is found to be 1.1 K for the $H_2O$ bands and 0.5 K for the $CO_2$ bands. Analysis of MODIS radiance comparisons with respect to IASI over six years found that the water vapor bands showed scatter up to 1.0 K in the H2O bands and 0.5 K in the CO2 bands (Moeller et al., 2014). The fusion comparison results are similar. Thus, it can be summarized that an order of magnitude spatial resolution (from 14 km to 750 m) has been added at the cost of introducing measurement offsets of 0.25 to 0.5 K and noise of 0.5 to 1.0 K. Results for all twelve months in 2018 (not shown) are similar; in fact, results for the entire S-NPP archive show comparably positive fusion results.

**(a)**

**(b)**

**Figure 2. Comparison of collocated MODIS and VIIRS/CrIS fusion water vapor (a) and carbon dioxide (b) band brightness**
**temperatures for the month of April 2018. Each data point in the plots (within a ten-degree brightness temperature bin)**
**represents more than ten thousand colocations.**

**3.      TPW and UTH Algorithm and Results**

Our retrieval of TPW and UTH from selected IR measurements adopts a statistical regression algorithm (Seemann et

al. 2003 and 2008; Li et al., 2000; Smith and Woolf, 1988; Hayden 1988) performed using clear sky radiances (and

BTs) measured over land and ocean for both day and night. The regression is developed with the SeeBor training

database (Borbas et al., 2005) that consists of over 15,000 atmospheric profiles globally and seasonally well

distributed. The water vapor retrieval algorithm has two parts, first the regression coefficients are calculated using

radiative transfer calculations, and then the regression retrieval is performed. The radiative transfer calculation of the

MODIS-like radiances of bands 25, 27, 28 and 30-36 is performed using the forward model called Radiative Transfer

for TOVS (RTTOV) Version 12 (Saunders et al., 2018). The regression relationships between the calculated BTs and retrieved moisture products are generated for four (and three) different BT zones over land and ocean, respectively, and 60 sensor zenith angles from nadir to 60°. The only other ancillary information required is the surface pressure, which is provided by NCEP Reanalysis data (Saha et al., 2010). TPW and UTH are determined for clear sky radiances measured by VIIRS and calculated from VIIRS+CrIS fusion. The retrieval approach is similar to that adopted for MODIS. There is a strong reliance on radiances from 6.7, 11, and 12 µm. The operational VIIRS cloud mask (called CLDMSK_L2_VIIRS_SNPP.001, Ackerman et al., 2019) is applied to VIIRS to characterize the probability of cloud cover.

Figure 3 shows CrIS TPW and UTH at the sounder FOV resolution; they outline the tropospheric moisture gradients at coarse (~14km) resolution for clear and partly cloudy skies. The soundings are obtained using the Dual Regression method (Smith et al. 2012, Weisz et al., 2013) which is a computationally fast, physically-based method that retrieves profiles as well as surface and cloud properties from high spectral resolution radiances measured in both clear and cloudy-sky conditions at single FOV resolution. TPW represents the total column integration of the moisture profile while UTH is the integration from 400 hPa to the top of the atmosphere. Also shown are the regression retrieval results for the VIIRS+CrIS fusion spectral band radiances (created using the MODIS-like IR spectral response functions) at higher spatial resolution (~750m) in pixels deemed to be clear in the VIIRS cloud mask. They display more refined features and improve the coverage, but show higher values of TPW off the coast of Baja and miss some of the UTH features in Wyoming and Colorado suggested in the CrIS soundings. While the results are derived from two independent algorithms, this example illustrates a challenge for the VIIRS split window search for nearby FOVs; the search will rely primarily on low level temperature and moisture features and less on mid to upper level moisture gradients.

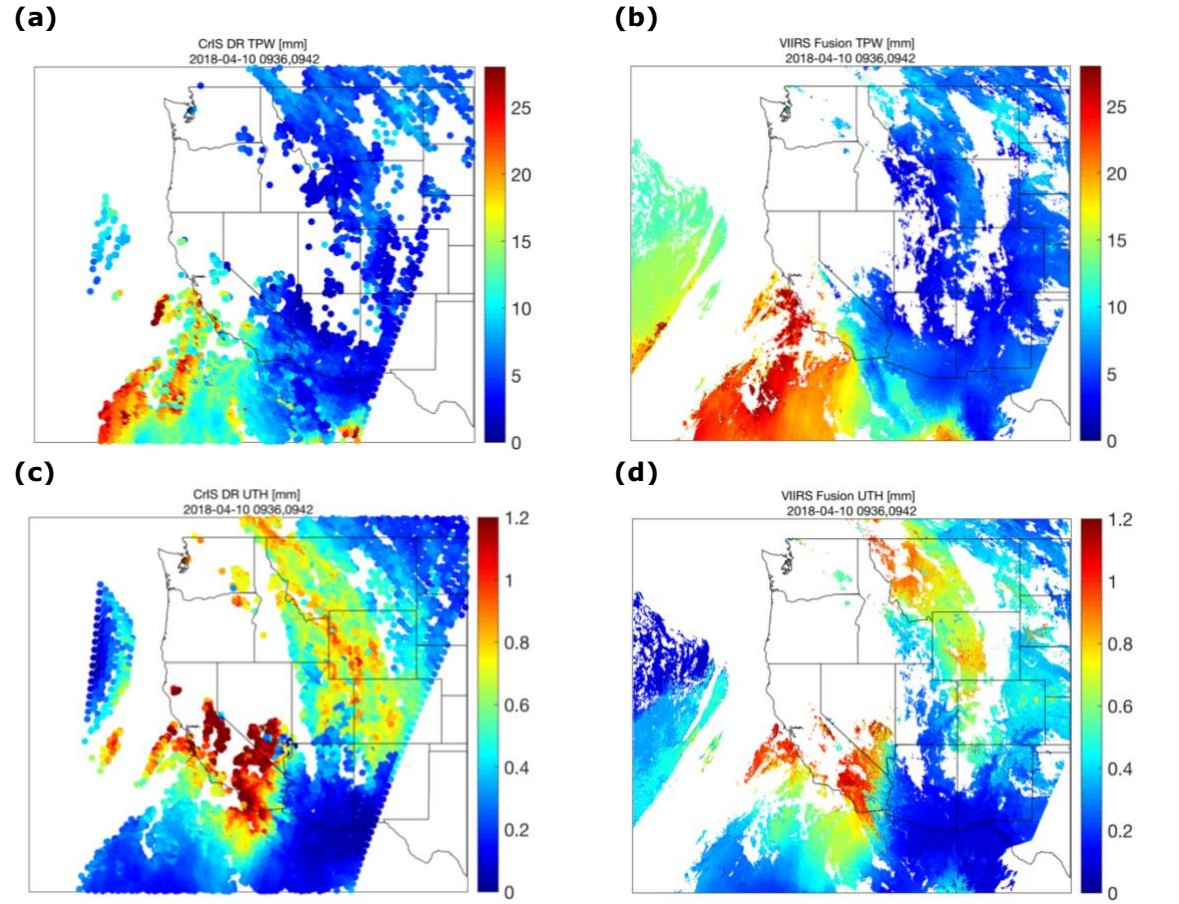

**(a)**

CrIS DR TPW [mm]
2018-04-10 0936,0942

**(b)**

VIIRS Fusion TPW [mm]
2018-04-10 0936,0942

**(c)**

CrIS DR UTH [mm]
2018-04-10 0936,0942

**(d)**

VIIRS Fusion UTH [mm]
2018-04-10 0936,0942

**Figure 3. TPW (a,b) and UTH (c,d) (both in mm) are shown for CrIS DR retrievals at sounder resolution (a,c) along with regression retrievals derived from VIIRS+CrIS fusion radiances at imager resolution (b,d) for 10 April 2018, at 0936 and 0942 UTC (CrIS granule start times).**

**3a. TPW Results**

A one-day evaluation of the VIIRS+CrIS TPW fusion product is shown in Fig. 4. Global comparisons for 9 April 2018 are made for the TPW field derived from (1) VIIRS+CrIS fusion radiances using the operational MODIS L2 algorithm, (2) MODIS operational Col 6.1 MYD08 (MYD08_D3.006, Platnick et al., 2015), (3) VIIRS-only (Borbas et al., 2019d), and (4) the VIIRS+NUCAPS (Borbas et al., 2019d) operational products developed under a NASA-funded project.

The VIIRS-only product is a statistical regression based on the split window radiances; it suffers from no information about mid- to upper tropospheric moisture. In the VIIRS+NUCAPS operational products, VIIRS IR measurements are merged with CrIS and ATMS water vapor soundings in an earlier attempt to continue the depiction of global moisture at high spatial resolution started with MODIS. A clear sky regression relationship has been established between TPW and VIIRS IR window BTs and NUCAPS water vapor soundings calculated from a global training radiosonde-based profile data set. NUCAPS TPW was added in clear and partly cloudy regions to enhance the TPW

depition and to extend the coverage. The CrIS and ATMS sounding products are provided by the NOAA Unique Combined Atmospheric Processing System (NUCAPS, Gambacorta, 2013). The main idea of merging these products is to capitalize on the unique strengths of each product's spatial and spectral characteristics in the infrared region. VIIRS, with solely the IR window channels, only gives some indication of low-level moisture (which constitutes much of the total column amount) and we complement this with CrIS+ATMS sounding column moisture retrievals. This VIIRS+NUCAPS algorithm follows the approach used for MODIS. A clear sky regression relationship is established between TPW (predictand), and VIIRS IR window brightness temperatures (BTs) and the NUCAPS TPW soundings (predictors) calculated from a global training radiosonde-based profile data set. To help differentiate surface emission and atmospheric moisture absorption and to get better surface characteristics in the forward model calculation, surface emissivity for the VIIRS channels used in the regression method has been assigned for each profile in the training dataset from the University of Wisconsin high spatial resolution surface emissivity database (Borbas et al, 2018). NUCAPS is added in clear and partly cloudy regions to enhance the TPW depiction and to extend the spatial coverage. First, the VIIRS-only clear-sky TPW is generated and stored; subsequently the VIIRS+NUCAPS TPW is calculated in clear-sky conditions. Gaps in the VIIRS+NUCAPS TPW field are filled with adjusted VIIRS-only or adjusted NUCAPS-only products. In this paper we use both the VIIRS-only and VIIRS+NUCAPS total column properties for evaluation.

Figure 4 shows that the global mean of the TPW derived from the VIIRS+CrIS fusion radiances is found to be 0.3 mm too low with a scatter of 3.3 mm when compared to the MYD08 TPW. The VIIRS-only operational TPW are 1.3 mm higher than the MYD08 TPW with a scatter of 4.0 mm; much of the VIIRS over-estimation of TPW occurs in the tropical oceans. VIIRS+NUCAPS TPW also compare well with the same 0.3 mm bias, but with a slightly higher 3.5 mm scatter in the comparison to MYD08 TPW. However, it does not capture the maxima in the Brazilian rainforest moisture found in the MYD08 TPW. The latitudinal distribution of the differences on Fig. 5 shows good agreement between VIIRS+CrIS fusion and MYD08 and over-estimation of VIIRS-only from 40˚ S to 10˚ N latitude. Over the tropics, where the highest moisture levels occur, the VIIRS+CrIS fusion product agrees more closely with the MYD08 than the VIIRS+NUCAPS, which mostly underestimates the water vapor content. For this one-day global comparison, providing $H_2O$ fields from fusion bands can be regarded as a success for bringing VIIRS+CrIS TPW into family with MYD08 TPW with a slightly better agreement than with the VIIRS+NUCAPS product, and additionally providing a significant improvement over the VIIRS-only product.

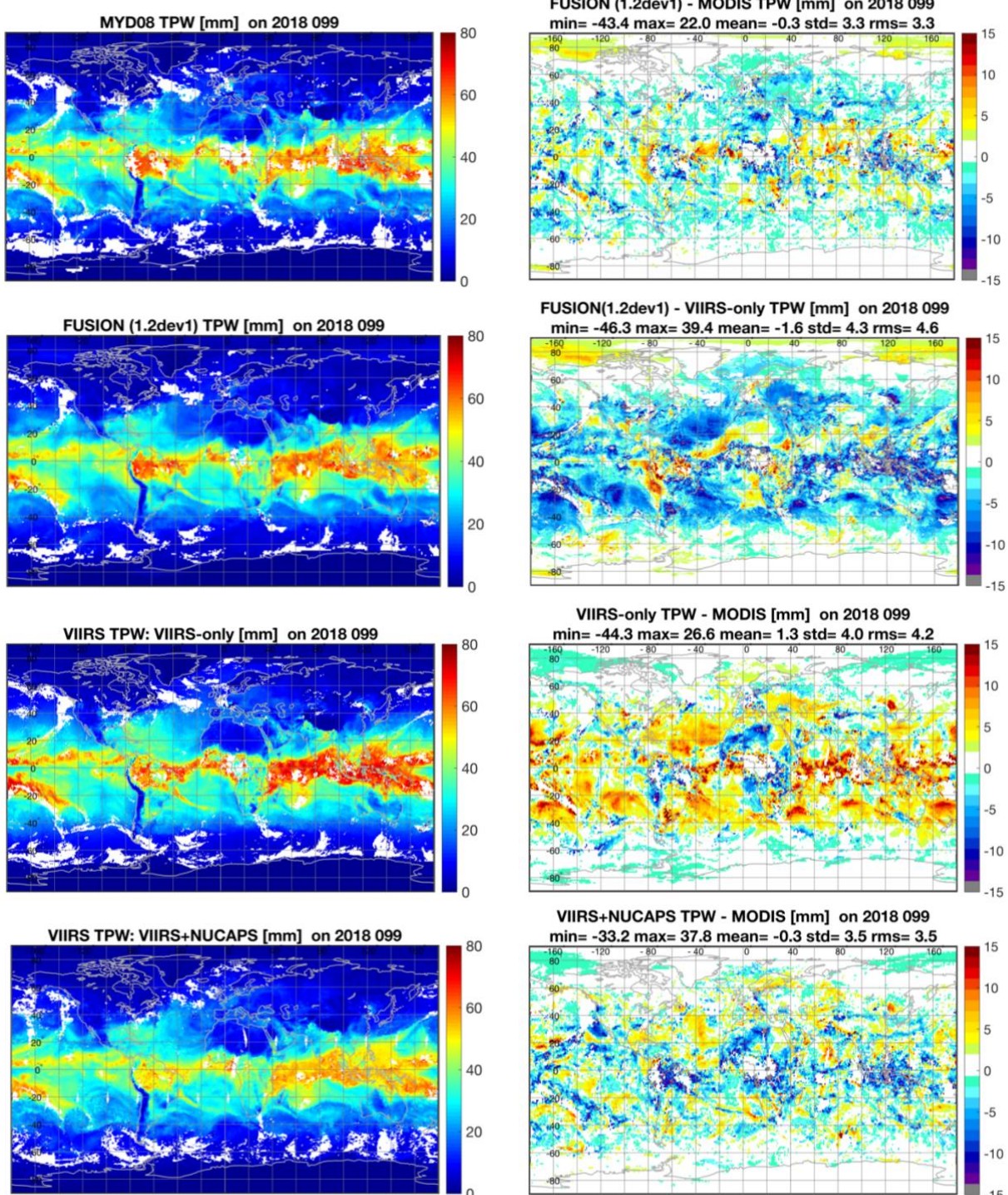

Figure 4. Left: geographical distribution of TPW [mm] results derived from the MODIS MYD08_D3 Collect 6 (left top), VIIRS/CrIS fusion (left second), VIIRS-only (left third) and the VIIRS+NUCAPS (bottom) products for 9 April 2018. The right panels show the corresponding difference fields with their statistics, such as, the minimum (min), maximum (max), mean, standard deviation (std) and root mean square (rms) differences are included in the subtitles.

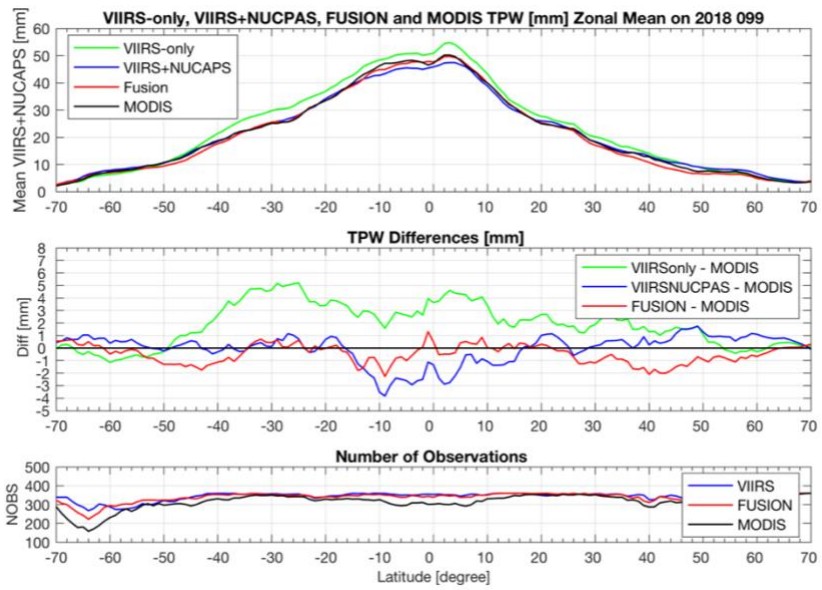

**Figure 5. Top: latitudinal distribution of TPW [mm] results for the same day and products as on Figure 4. The middle panel illustrates the corresponding differences while the third panel shows the number of observations occurred in each one-degree latitude bins.**

The one-day comparisons are now extended to monthly comparisons. Figure 6 shows zonal scatter plots for the month of April 2017 of VIIRS+CrIS fusion, VIIRS-only, and VIIRS+NUCAPS TPW, each with respect to MYD08. The segmentation is into three zones of 60° to 30° N (northern mid-latitudes), 30° N to 30° S (tropics), and 30° to 60° S latitudes (southern mid-latitudes). VIIRS+CrIS fusion TPW shows differences in all three zones in the mean (-1.16, 0.24, -0.49 mm respectively) and standard deviation (0.95, 1.89, and 0.66 mm respectively); the dry bias is greater than 1 mm in the northern mid latitudes and is pervasive in the eastern U.S., the northern Atlantic Ocean, through Europe, and continuing to western Russia. Overall good agreement is found in dry (less than 5 mm) as well as wet (greater than 60 mm) atmospheres. Similar comparisons are less favorable for VIIRS-only and VIIRS+NUCAPS, with the exception of northern mid-latitude where VIIRS+NUCAPS shows a smaller absolute bias in the mean of 0.92 mm. The VIIRS+CrIS fusion product compares with MODIS TPW within the MODIS product accuracy as determined from CART site comparisons (Borbas et al., 2011, MODIS Atmospheric Products ATBD), thus VIIRS+CrIS is shown to be a viable source for the MODIS moisture product record continuation.

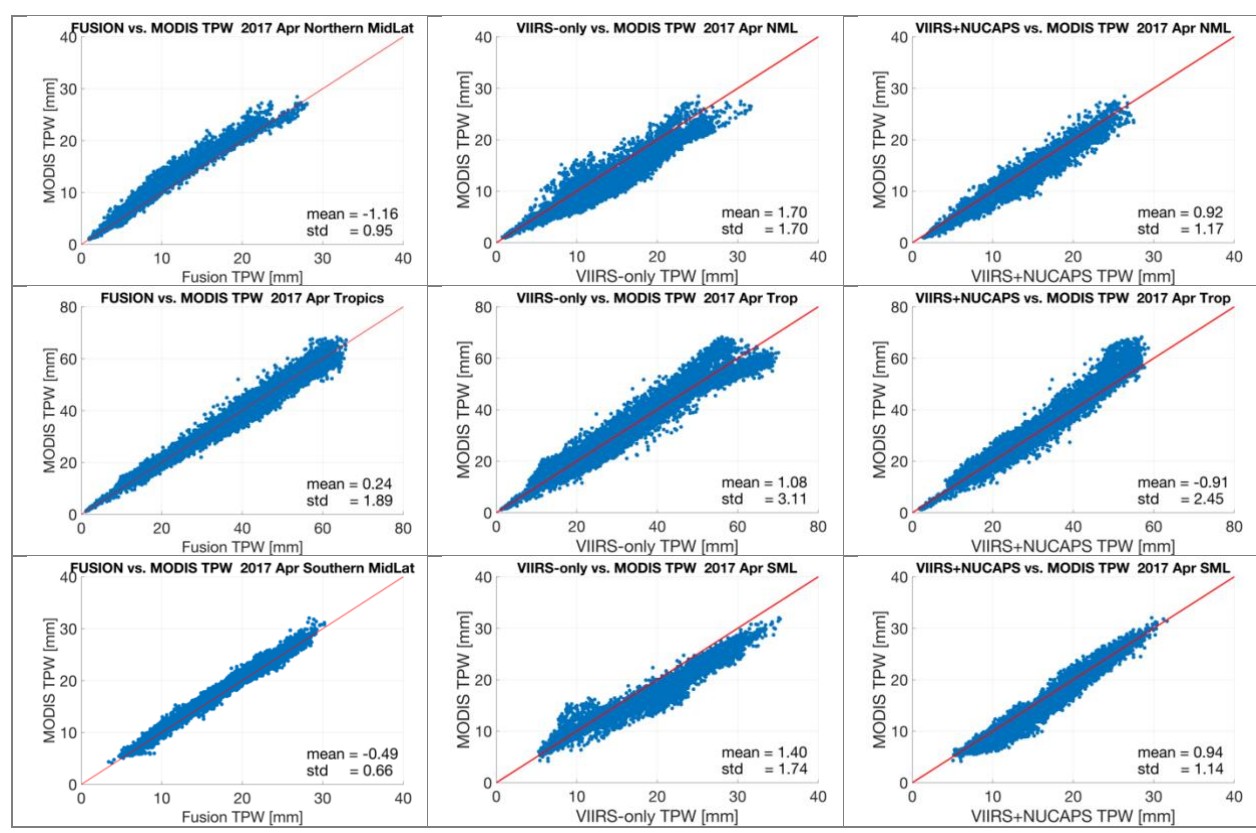

**Figure 6. TPW scatter plot of VIIRS+CrIS fusion (left), VIIRS-only (middle) and VIIRS+NUCAPS (right) versus MODIS MYD08_M3 Col6.1 for Northern mid latitudes between 30° and 60° N (top), Tropics between 30°S and 30° N(middle), and Southern mid latitudes between 30° and 60°S (bottom) on April 2017.**

Figures 7 and 8 show the global comparison of monthly differences for January 2017. These results reinforce the one day results, especially with regard to VIIRS+CrIS TPW being the best match of the three VIIRS derived TPWs with MYD08 TPW; VIIRS+CrIS has the lowest mean difference at 0.2 mm and standard deviation of 1.4 mm compared to respectively 1.1 and 2.7 mm for VIIRS alone and 0.3 and 2.0 mm for VIIRS+NUCAPS. The improvement is most noticeable (see Fig. 8) over the Brazilian rain forest and the ITCZ (Inter Tropical Convergence Zone).

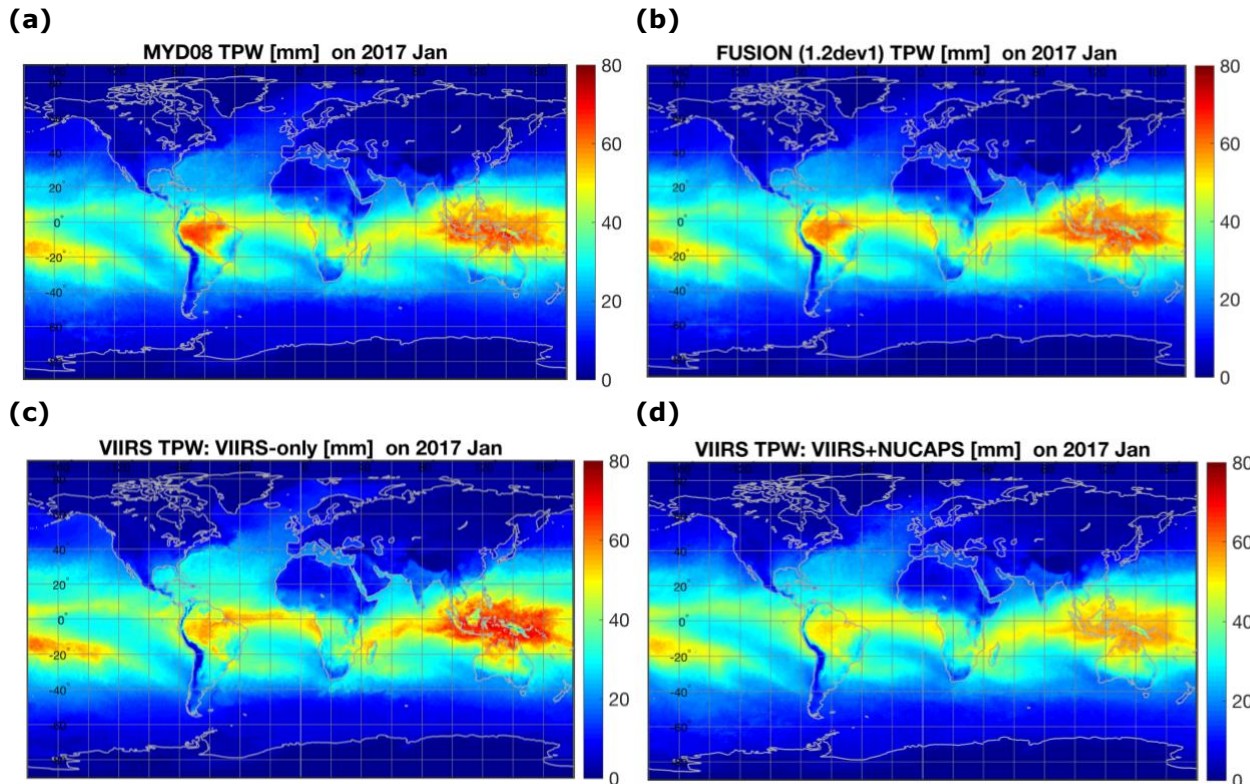

**Figure 7. January 2017 geographical distribution of TPW [mm] results derived from the (a) MODIS MYD08_M3 Collect 6.1, (b) VIIRS/CrIS fusion, (c) VIIRS-only, and (d) VIIRS+NUCAPS.**

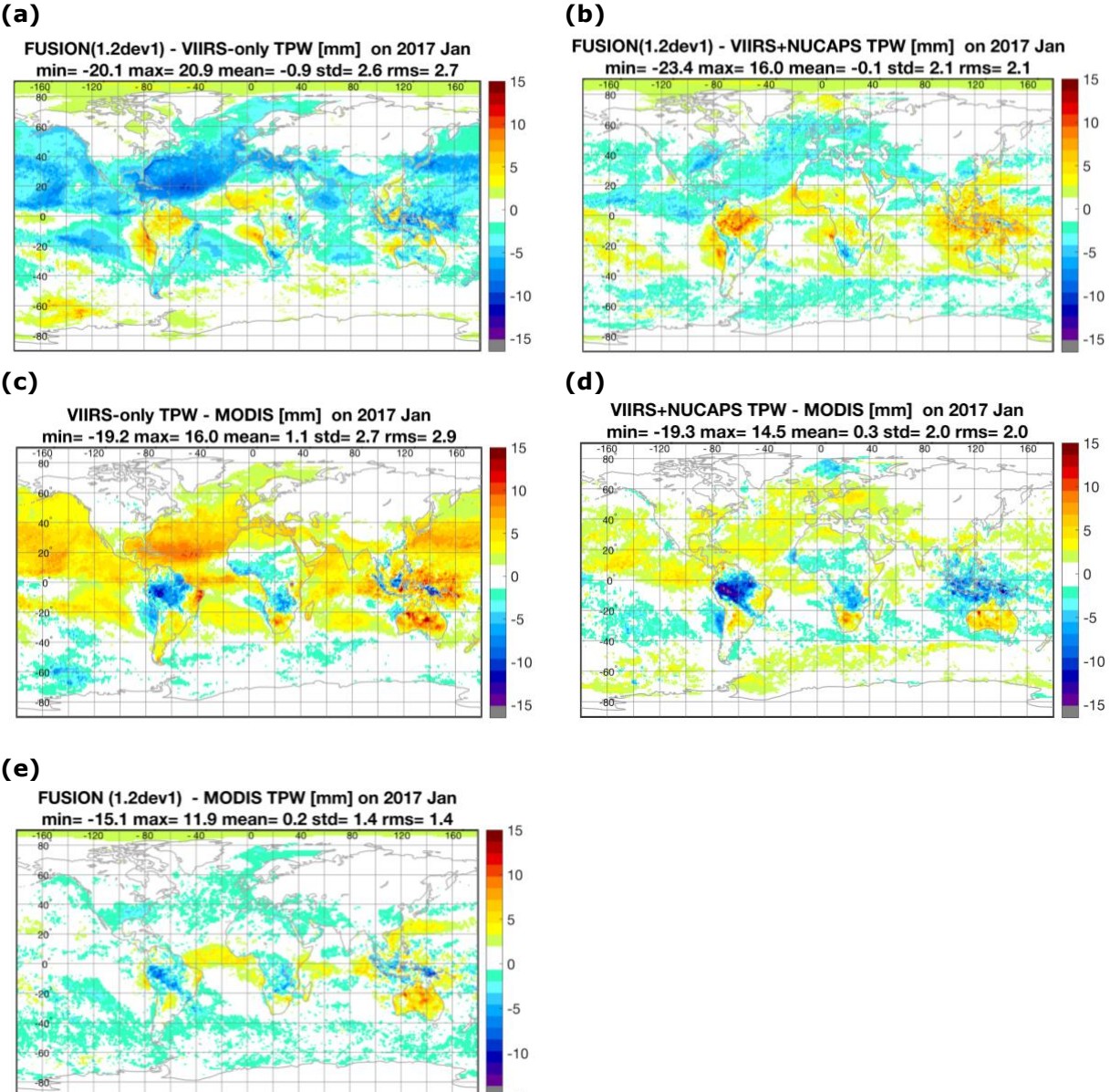

**Figure 8. January 2017 TPW [mm] difference fields of VIIRS+CrIS fusion minus VIIRS-only (a), VIIRS+CrIS fusion minus VIIRS+NUCAPS (b), VIIRS-only minus MODIS (c), VIIRS+NUCAPS minus MODIS (d), and VIIRS/CrIS fusion minus MODIS. MODIS refers to MYD08_M3 C6.1 products. Minimum (min), maximum (max), mean, standard deviation (std) and root mean square (rms) differences are shown in the title of each panel.**

To extend this analysis to a four season evaluation, VIIRS+CrIS TPW differences with respect to MYD08 TPW are shown for January, April, July, and October 2017 in Fig. 9. Mean agreement ranges from 0.0 mm in April and 0.4 mm in October; the standard deviation is largest in July at 1.8 mm, which is still smaller than the standard deviation of any of the other three VIIRS-derived products in January 2017. Local VIIRS+CrIS overestimations occur over Australian deserts in January and during the Indian monsoon in July; underestimation is found in the Brazilian rainforest and the ITCZ in January and the Saharan desert in July. In over 300 colocations, MYD08 TPW were compared to ground- based microwave radiometer determinations retrievals of TPW and found to be dry biased by

1 ~~0.9 mm with a rms of 3.2 mm (Borbas et al., 2011).~~ Overall VIIRS+CrIS TPW agrees very well with MODIS TPW

2 for all four months representing the four seasons.

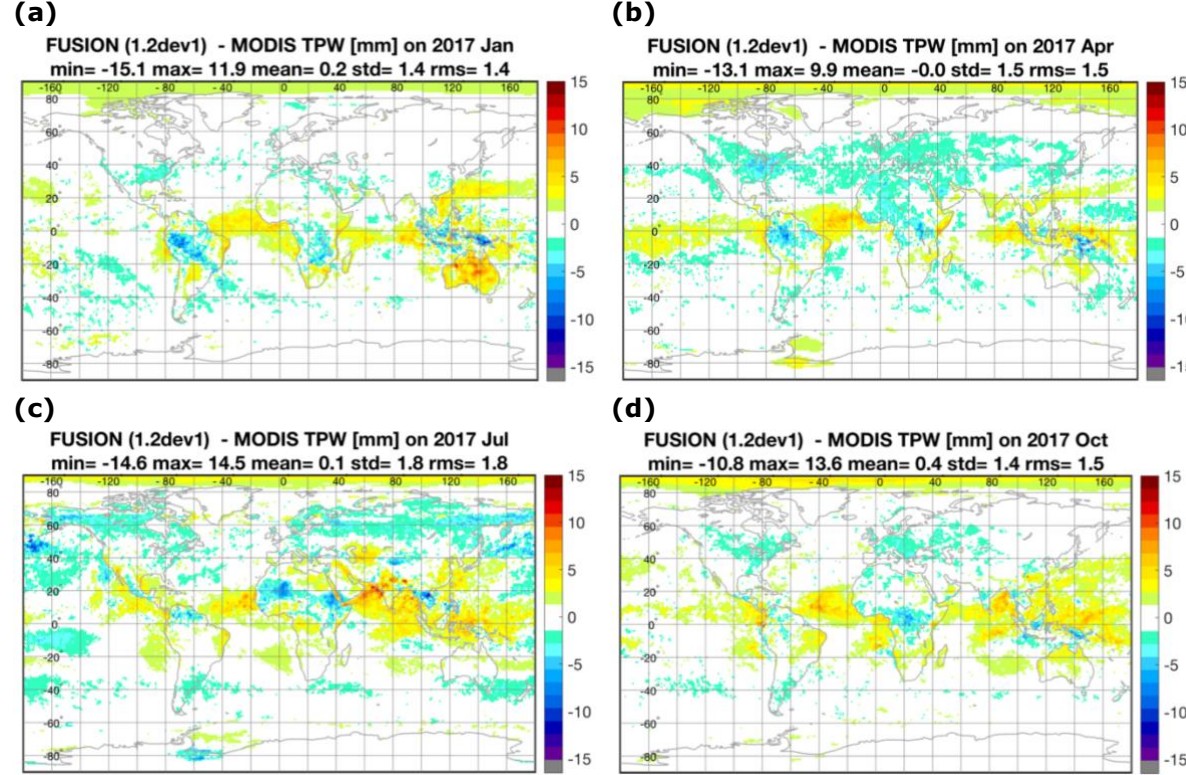

**Figure 9. Geographical distribution of TPW [mm] differences between the VIIRS+CrIS fusion and the MODIS MYD08_M3 Collect 6.1 products for (a) January, (b) April, (c) July, and (d) October 2017 representing the four seasons. Minimum (min), maximum (max), mean, standard deviation (std) and root mean square (rms) differences are shown in the title of each panel.**

**3b. UTH Results**

Figure 10 shows the results for the VIIRS+CrIS fusion UTH product. The UTH global images on the top two panels

show the spatial distribution of UTH within the 0-3 mm range. Here the global mean derived from the VIIRS+CrIS

fusion radiances is found to be 0.02 mm higher with a scatter of 0.14 mm when compared to the MYD08 UTH. Local

differences of ±1 mm are found in the tropics. The latitudinal distribution of the differences (Fig. 11) shows modest

overestimation in the VIIRS+CrIS fusion UTH everywhere with a peak from 10˚S to the equator. Note that the

operational VIIRS moisture products do not currently include the UTH product, but only total column moisture

information, since VIIRS has a limited ability to sense the upper tropospheric moisture. Again, in this global

comparison for one day, use of the $H_2O$ fusion bands brings VIIRS+CrIS fusion UTH into family with MYD08 UTH.

Without the fusion radiances, VIIRS has little or no sensitivity to UTH.

(a)

(b)

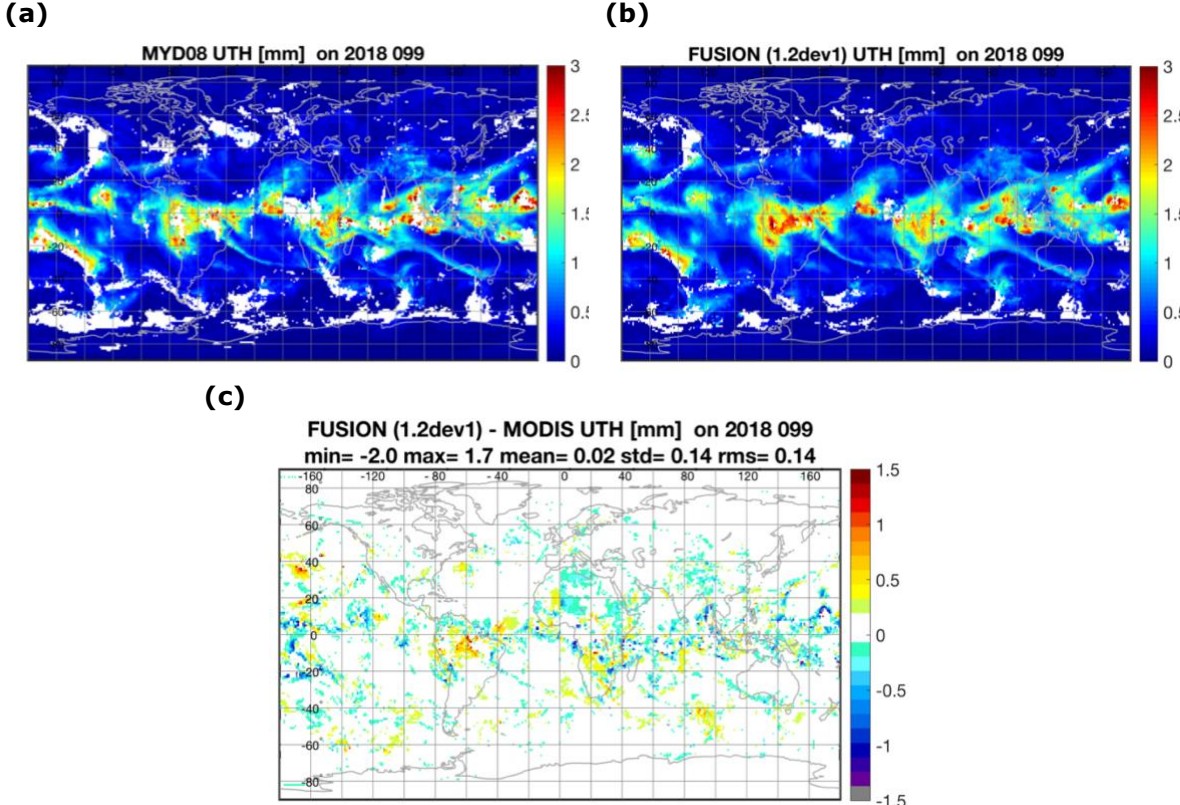

(c)

Figure 10. Geographical distribution of UTH [mm] results derived from the MODIS Collect 6 (a), and VIIRS/CrIS fusion (b), and their difference (c) for 9 April 2018. The minimum (min), maximum (max), mean, standard deviation (std) and root mean square (rms) differences are shown in the title **of panel (c).**

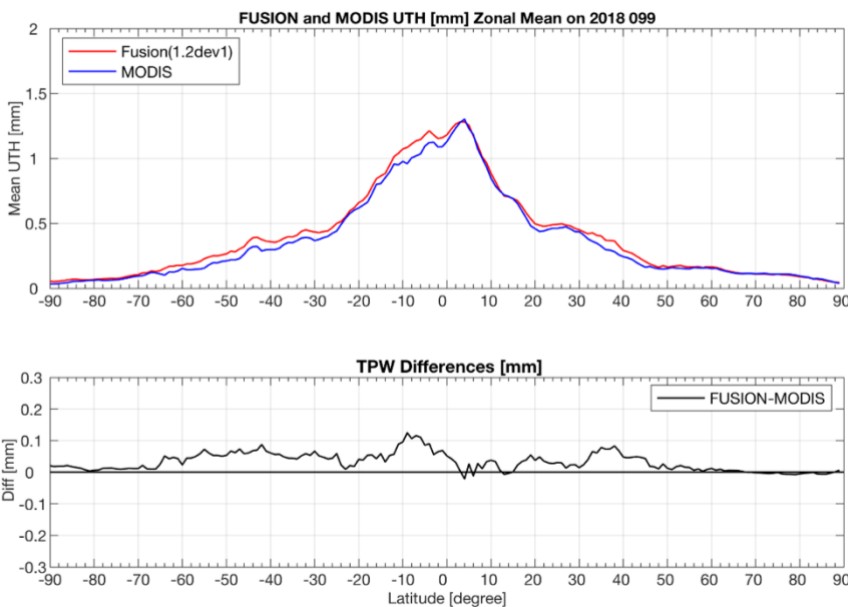

Figure 11. Top: latitudinal distribution of UTH [mm] results for MODIS and VIIRS+CrIS Fusion derived from the same 9 April 2018 data shown in Figure 10. The bottom panel illustrates the corresponding differences. The number of observations found in each one-degree latitude bins is shown in the bottom panel of Figure 5.

Figure 12 shows the UTH comparison results for one month in each season that complement the TPW results in Fig.

9.  Mean agreement for VIIRS+CrIS UTH with MODIS UTH ranges from 0.03 to 0.56 mm and standard deviation

from 0.05 to 0.08 mm.  Greatest local differences are found with VIIRS+CrIS too wet in the ITCZ in January, too wet

in the Brazilian rain forest in April, too dry in the Himalayas and too wet in India in July, and again too wet in the

ITCZ in October. Overall, the results are typically within 10% of each other and accurate enough to determine daily

and seasonal variability.

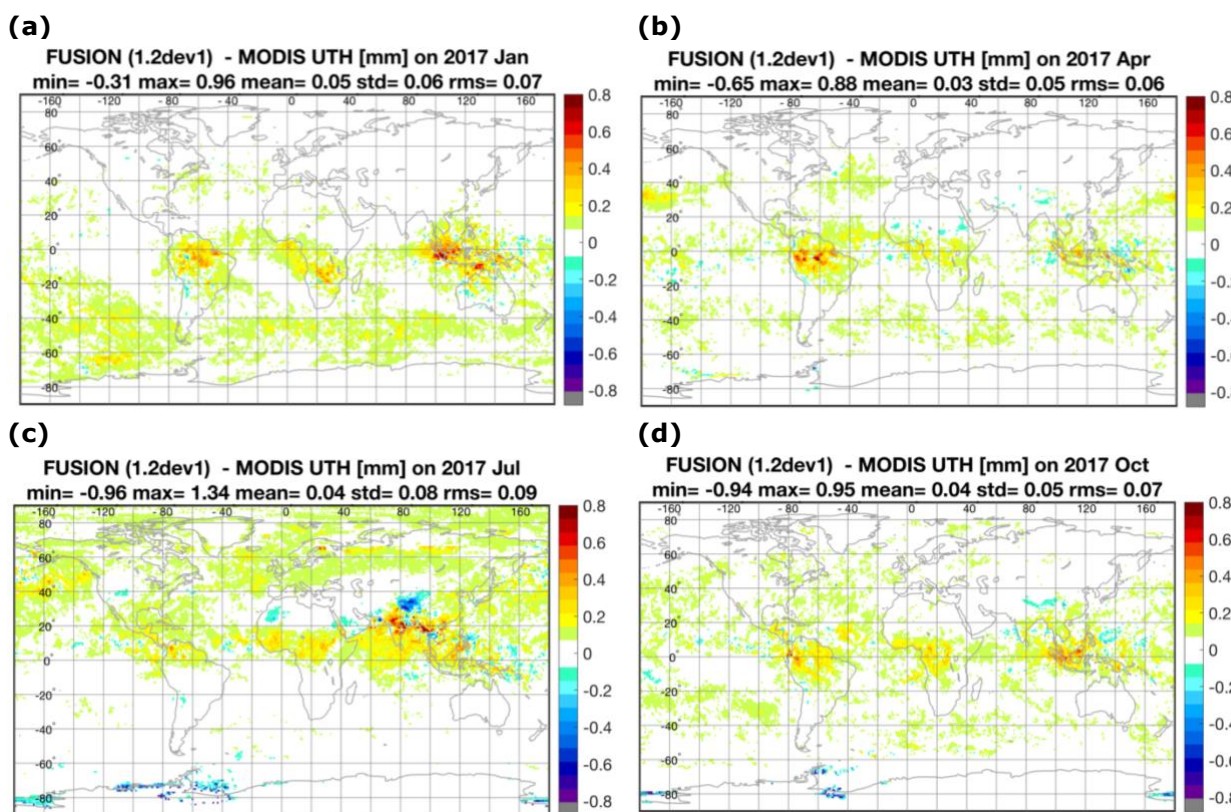

**Figure 12. Same as Figure 9, but for UTH.**

**4.        Summary and Conclusions**

The absence of water vapor and $CO_2$ absorption IR spectral bands on the VIIRS imager on the Suomi-NPP and NOAA-

20 polar-orbiting platforms limits the capability for tropospheric moisture retrievals, especially for upper tropospheric

moisture. This study shows the advantage of using IR absorption bands 4.5, 6.7, 7.3,13.3, 13.6, 13.9, and 14.2 µm that

are constructed at VIIRS spatial resolution (750m) using a data fusion approach using both sounder (CrIS) and imager

(VIIRS) measurements following the approach in Weisz et al. (2017). The positive impact of adding the constructed

fusion spectral bands on TPW and UTH retrievals is demonstrated. The moisture retrievals are based on the MODIS

MYD07 Collection 6.1 algorithm package. Evaluation of the resulting moisture products are performed through

comparisons to the operational MODIS Collection 6.1 and VIIRS (VIIRS-only and VIIRS+NUCAPS) version 1.0

moisture products.

Improvements in VIIRS+CrIS products, enabled by addition of fusion radiances, over the VIIRS-only and VIIRS+NUCAPS products are observed for TPW when quantitatively compared to the MYD08 products. In our one month study for January 2017, the global mean of the TPW derived from the VIIRS+CrIS fusion radiances is 0.2 mm higher with a scatter of 1.4 mm when compared to the MYD08 TPW; without the fusion radiances (VIIRS-only product) the mean is 1.1 mm too high with a scatter of 2.7 mm with most of the over-estimation occurring in the tropics. The VIIRS+CrIS fusion TPW also demonstrates improvement over the VIIRS+NUCAPS TPW (with 0.3 mm mean and 2.0 mm scatter with respect to the MYD08 product). Similar TPW results are also found for one month in each season of 2017. VIIRS+CrIS UTH, now possible with the addition of the fusion radiances, is found to be within 10% of the MYD08 UTH in mean and scatter for the same four months.

The results in this study are limited to a VIIRS sensor scan angle of 50° to minimize the impact of the CrIS swath being less than that of the imager. These findings are limited in scope but clearly demonstrate the potential in the use of the fusion IR absorption spectral bands in generating moisture products and continuing the moisture record from MODIS and the previous generations of polar orbiting satellite sensors. In future work, we plan to extend this evaluation to longer time periods, undertake a global comparison of VIIRS+CrIS fusion moisture products with ground-based measurements (Bedka et al 2010, Roman et al 2016), and potentially replace the operational VIIRS+NUCAPS moisture products with the VIIRS+CrIS fusion derived moisture products.

**Data availability**. The VIIRS/SNPP Cloud Mask, fusion, water vapor (WATVP) products and the Level-3 MODIS MYD08 products used in this study can be obtained from the NASA Level1 and Atmosphere Archive & Distribution System (LAADS) Distributed Active Archive Center (DAAC), Goddard Space Flight Center, USA. (https://ladsweb.modaps.eosdis.nasa.gov/search/ )

**Competing Interests**. The authors declare that they have no conflict of interest.

**Author Contributions**. E. Eva Borbas conceived and designed the TPW regression method, conducted the impact study and performed the analyses. Chris Moeller performed the fusion radiance validation in Sect.2. W. Paul Menzel and Bryan A. Baum made critical suggestions on the design of the study and significant improvements to the manuscript. Elisabeth Weisz provided expertise on the use of fusion products.

**Acknowledgements**

The authors gratefully acknowledge support from NASA grants 80NSSC18K0816 and 80NSSC18K0816. We are grateful for the encouragement and support by Dr. Hal Maring (NASA Headquarters, Washington, DC). The fusion data are generated by the Atmosphere SIPS at University of Wisconsin – Madison and sent to LAADS for public distribution. The writing of this paper benefited from discussions with our colleague Richard Frey for his insight with the VIIRS cloud mask. We thank Pascal Brunel (Meteo-France) for providing the spectrally shifted MODIS

coefficients for RTTOV, Geoff Cureton and Ethan Nelson for their efforts at the A-SIPS and Bhaskar Ramachandran for help in staging the fusion product at LAADS.

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
