# Peer review of "Improvement in tropospheric moisture retrievals from VIIRS"

_Atmospheric Measurement Techniques, 2020_

## Referee Comment (RC1) · Anonymous Referee #1 · 17 Aug 2020

Review of "Improvement in tropospheric moisture retrievals from VIIRS through the use of infrared absorption bands constructed from VIIRS and CrIS data fusion" manuscript, by E. Borbas et al.

The manuscript presents a novel way to retrieve humidity columns from the spatially highly resolved VIIRS imaging instrument with the hyperspectral sounder CrIS at high spatial sampling and hence enable continuity with the MODIS products. It is found overall well structured and well written, with clear and concise discussions and appropriate choice of illustrations. The study is interesting in particular in the perspective of

climate studies, to enable continuous records of humidity products originating from different programmes. The manuscript is understood as a demonstration of the potential of the method, which could be further elaborated and validated in a second step. The manuscript is recommended for publication with minor revisions, addressing the few general and specific suggestions below.

General:

The introduction and conclusion should clarify the scope of this paper, i.e. to test a first attempt to aim continuity with MODIS humidity products by attempting retrievals from data fusion. But with further work needed for validation (e.g. longer periods, use of independent reference measurements...) and possibly for the retrieval method itself. It is surprising that IR+MW retrievals (NUCAPS) gives so large dry biases compared to the other products. Because it is the only product exploiting microwave measurements, one would expect good precision here. Can the authors discuss these observations a little bit? In particular drawing from respective validation papers and reports published in the past. Is it envisaged at some stage to perform retrievals from multi-spectral data fusion: VIIRS+CrIS+ATMS? I.e. to try and exploit the maximum information content and enable production in cloudy conditions too. A line or two to comment on this, e.g. in the conclusion/outlook or a brief discussion in what no added-value may be expected would give useful perspective to the reader.

Specific:

P4.L13-14 /Fig.2: clarify if the "mean clear-sky BTDs" is a bias (=mean) or a standard deviation ("increasing measurement noise" of L17). If mean=bias, then what is the actual dispersion, which is the real noise increase.

P5.L8: not the object of this paper, but 15000 training patterns seems rather small in common ML approaches nowadays. Can the authors comment about possible limitations with this approach?

P8.L16: "also performs well ... but ..." I suggest avoiding to say that a product performs well on the reason that it compares better to MODIS. In particular here, we should expect that the utilisation of microwave together with hyperspectral infrared sounders offers higher capabilities for accurate TPW characterisation than the MODIS imager on its own. Especially in very moist to cloudy atmospheres. Sampling effects of CrIS/ATMS vs MODIS resolution may however explain small dry bias in moist atmospheres and conversely. It would be instructive to compute spatial averaging of MODIS products at CrIS spatial resolution to see if the dry bias on the moistest atmospheres remains.

A comparison to e.g. ground-based TPW sensing (Bedka et al. 2010 or Roman et al. 2016) is advisable to be able to conclude on the final respective skills of the fusion and the single products. It is however an interesting results to match closely MODIS retrievals with the fusion products, which can be highlighted as such.

P13.L12 - P14.L1-2: the link between the former results (2011...) and the present study and the conclusion asserted is not clear. The argument would deserve some elaboration. VIIRS+CrIS TPW could agree very well with MODIS TPW even in the absence of a 3rd reference dataset. However it would be difficult to infer which one is closer to the truth.

P14.L12 and P16/L16: clarify if MODIS UTH is really MYD07 or indeed MYD08 as per manuscript introduction. If MYD07, some more details of that version vs MYD08 would be useful to the readu to interpret the results.

The conclusion should open to validation against fiducial reference measurements to conclude on the differences between MODIS and the fusion products, or explain in what this is judged not necessary, if so.
* * *

---

## Referee Comment (RC2) · Anonymous Referee #2 · 28 Sep 2020

The manuscript titled: "1. Improvement in tropospheric moisture retrievals from VIIRS through the use of infrared absorption bands constructed from VIIRS and CrIS data fusion" by E. Borbas et al. reads well, the methodology is sound and the results are clearly explained. I strongly suggest that this manuscript is accepted for publication, pending minor revisions outlined below.

To make the manuscript science question more urgent, I would like to suggest that the authors added more text in the introduction to describe why this data fusion product is important. For example, could the authors say a few more words on the need for such a

[Figure]

data fusion product rather than just using water vapor estimates from the CrIS sensor? Can they provide a reference to similar existing products from MODIS in support of the applicability and/or user request of this product? The author could simply add a few sentences on the benefit of TPW estimates at a high spatial resolution (750m) versus the coarser spatial resolution of the CrIS sensor and state why high spatial resolution TPW is important for end- users' applications. Continuity of the MODIS data record is also important, but the authors only mention it in the conclusion remarks. It would be useful to state it upfront, in the introduction section as well. This is a minor addition but would make the paper a lot more relevant in the framework of TPW near real time or long-term applications.

Page 2, line 2. "estimates" should replace "determination" Page 3, line 2: a definition of "split-windows" could help non-expert readers. Page 3, line 2: what is a k-d tree search algorithm?

Was the data fusion technique applied to clear sky only pixels or all-sky scenes? Page 4, line 10 says: "the scene must be high confidence clear" Is it just the way the validation was done, that is a clear-sky only validation? Same question for the scan angles: "must be less than 50 degrees". Or is it because the data fusion technique only applies to clear-sky, less then 50 degrees pixels?

Figure 2 (a) and (b). What do these differences mean? Can the author provide a comparison, on the same figures, with respect to the instrument noise of the VIIRS and MODIS instrument?

Page 8, line 3: This sentence: "A clear sky regression relationship is established between TPW and VIIRS IR window brightness temperatures (BTs) and the NUCAPS TPW soundings calculated from a global training radiosonde-based profile data set." might not be entirely not clear. What is the training ensemble, what are the predictors?

Page 8, line 5. Can the author provide more description on the use of the surface emissivity database, when they state: "A high spatial resolution surface emissivity database

(Borbas et al, 2018) is used to help differentiate surface emission and atmospheric moisture absorption."

Page 16, line 10: "CO2 absorption IR spectral bands" is this part of the sentence necessary to the extent of moisture retrieval products?

A general comment about Figure 6. The VIIRS+CrIS product improves significantly over the VIIRS only and VIIRS+NUCAPS, in terms of both mean and sdv when compared to the MODIS product. Can the authors explain the impact of this improvement in terms of continuity of the data record. Are there specificities requirements? This remark would strengthen the value of figure 6 and, more importantly, the very final conclusion remark.

---

## Author Comment (AC1) · 23 Oct 2020

Responses to Anonymous Referee #1

General:

The introduction and conclusion should clarify the scope of this paper, i.e. to test a first attempt to aim continuity with MODIS humidity products by attempting retrievals from data fusion. But with further work needed for validation (e.g. longer periods, use of independent reference measurements...) and possibly for the retrieval method itself.

It is surprising that IR+MW retrievals (NUCAPS) gives so large dry biases compared to the other products. Because it is the only product exploiting microwave measurements, one would expect good precision here. Can the authors discuss these observations a little bit? In particular drawing from respective validation papers and reports published in the past. Is it envisaged at some stage to perform retrievals from multi-spectral data fusion: VIIRS+CrIS+ATMS? I.e. to try and exploit the maximum information content and enable production in cloudy conditions too. A line or two to comment on this, e.g. in the conclusion/outlook or a brief discussion in what no added-value may be expected would give useful perspective to the reader.

Response: The VIIRS+NUCAPS retrievals refer to a combination of VIIRS-only split window TPW retrievals supplemented by NUCAPS retrievals in cloudy regions. A clear-sky regression relationship is established between total precipitable water vapor (TPW), and VIIRS IR window brightness temperatures (BTs) and NUCAPS water vapor soundings calculated from a global training radiosonde-based profile data set. NUCAPS TPW is added in clear and partly cloudy regions to enhance the TPW depiction and to extend the coverage. It is not the NUCAPS retrieval. We intend to evaluate the utility of the ATMS data in future work.

Specific:

P4.L13-14 /Fig.2: clarify if the "mean clear-sky BTDs" is a bias (=mean) or a standard deviation ("increasing measurement noise" of L17). If mean=bias, then what is the actual dispersion, which is the real noise increase.

Response: RMS values are found to be within 1.1 K for the water vapor bands and within 0.5 K for the $CO_2$ bands. Text now indicates this.

P5.L8: not the object of this paper, but 15000 training patterns seems rather small in common ML approaches nowadays. Can the authors comment about possible limitations with this approach?

Response: Our goal is to work towards routine atmospheric retrievals of TPW and UTH from VIIRS+CrIS data. However, there is a really big difference in using a ML approach and something like our approach for routine operational use. If a problem develops in the software, one has to find it, evaluate it, and fix it quickly. With an AI or ML approach, this is much more difficult. To the question, the construction of 15,000 profiles is a major undertaking, and it is sufficient for training purposes. Having said this, it would be interesting to compare our results with those from a ML approach at some point.

P8.L16: "also performs well ... but ..." I suggest avoiding to say that a product performs well on the reason that it compares better to MODIS. In particular here, we should expect that the utilisation of microwave together with hyperspectral infrared sounders offers higher capabilities for accurate TPW characterisation than the MODIS imager on its own. Especially in very moist to cloudy atmospheres. Sampling effects of CrIS/ATMS vs MODIS resolution may however explain small dry bias in moist atmospheres and conversely. It would be instructive to compute spatial averaging of MODIS products at CrIS spatial resolution to see if the dry bias on the moistest atmospheres remains.

Response: Text has been changed to "compare well." We agree that it is a good idea to compute spatially averaged MODIS products at CrIS spatial resolution to see if the dry bias still exists. We will perform such a study in the future.

A comparison to e.g. ground-based TPW sensing (Bedka et al. 2010 or Roman et al. 2016) is advisable to be able to conclude on the final respective skills of the fusion and the single products. It is however an interesting results to match closely MODIS retrievals with the fusion products, which can be highlighted as such.

Response: The immediate goal of this work is to establish the feasibility of extending the MODIS derived TPW and UTH products into the future with VIIRS/CrIS fusion. The validation of MODIS products with respect to ground based measurements is implicit in the VIIRS/CrIS fusion comparisons with MODIS. Initial validation of ABI/CrIS fusion

products with ground-based measurements has begun with the newly-published work of Anheuser et al (2020); they compared soundings with radiosondes at the CART site. We intend to undertake a global comparison of VIIRS/CrIS moisture products with ground-based measurements in the future. A sentence has been added to the Summary and Conclusions section.

P13.L12 - P14.L1-2: the link between the former results (2011...) and the present study and the conclusion asserted is not clear. The argument would deserve some elaboration. VIIRS+CrIS TPW could agree very well with MODIS TPW even in the absence of a 3rd reference dataset. However it would be difficult to infer which one is closer to the truth.

Response: Please see previous answer. Comparison of VIIRS/CrIS fusion with MYD08 within earlier bias and rms with respect to radiosondes suggests the feasibility to continue MODIS TPW and UTH records into the future.

P14.L12 and P16/L16: clarify if MODIS UTH is really MYD07 or indeed MYD08 as per manuscript introduction. If MYD07, some more details of that version vs MYD08 would be useful to the reader to interpret the results.

Response: In the Introduction, we added this text: "The MYD07 is a Level-2 swath product that provides temperature and water vapor profiles at 5-km spatial resolution, while the MYD08 provides water vapor on an 8-day global grid at 1ËŽ x 1ËŽ resolution." Both MYD07 and MYD08 are used in this manuscript. In this particular place MYD08 was used. Thank you for pointing out this error.

The conclusion should open to validation against fiducial reference measurements to conclude on the differences between MODIS and the fusion products, or explain in what this is judged not necessary, if so.

Response: Closer comparison of VIIRS/CrIS fusion with MODIS results over VIIRS-only is the goal of this work. Validation against ground measurements is implicit in the

MODIS publications cited in the references. Explicit validation is planned and is stated in the text.

Please also note the supplement to this comment:
https://amt.copernicus.org/preprints/amt-2020-248/amt-2020-248-AC1-supplement.pdf

---

## Author Comment (AC2) · 23 Oct 2020

Responses to Anonymous Referee #2

The manuscript titled: "1. Improvement in tropospheric moisture retrievals from VIIRS through the use of infrared absorption bands constructed from VIIRS and CrIS data fusion" by E. Borbas et al. reads well, the methodology is sound and the results are clearly explained. I strongly suggest that this manuscript is accepted for publication, pending minor revisions outlined below.

[Figure]

To make the manuscript science question more urgent, I would like to suggest that the authors added more text in the introduction to describe why this data fusion product is important. For example, could the authors say a few more words on the need for such a data fusion product rather than just using water vapor estimates from the CrIS sensor? Can they provide a reference to similar existing products from MODIS in support of the applicability and/or user request of this product? The author could simply add a few sentences on the benefit of TPW estimates at a high spatial resolution (750m) versus the coarser spatial resolution of the CrIS sensor and state why high spatial resolution TPW is important for end- users' applications. Continuity of the MODIS data record is also important, but the authors only mention it in the conclusion remarks. It would be useful to state it upfront, in the introduction section as well. This is a minor addition but would make the paper a lot more relevant in the framework of TPW near real time or long-term applications.

Response: thank you for the suggestions. Given the positive results described in this article and continuing work, we plan to replace the operational VIIRS+NUCAPS moisture products with the fusion derived moisture products to be able to provide continuation of the MODIS MOD07 data record. The benefits associated with the continuation of such a high spatial resolution product include, for example, the observation of high spatial scale weather phenomena (weather forecasting), urban heat islands (Hu and Brunsell, 2015), and in determining atmospheric correction for high spatial resolution remote sensing products, such as the MODIS land surface temperature products (Proud et al, 2010, Hulley et al, 2017; Wen, 2010). This text has been added to the end of the Introduction section.

Page 2, line 2. "estimates" should replace "determination"

Response: The word "determination" was replaced by "retrieval" where it appeared in the manuscript.

Page 3, line 2: a definition of "split-windows" could help non-expert readers.

Response: Text has been added to explain the term "split window".

Page 3, line 2: what is a k-d tree search algorithm?

Response: k-d refers to multidimensional search, which is a standard routine in the computer science world. We included an appropriate reference to the text: J. L. Bentley, "Multidimensional binary search trees used for associative searching," Commun. ACM 18(9), 509–517 (1975), http://dx.doi.org/10.1145/361002.361007.

Was the data fusion technique applied to clear sky only pixels or all-sky scenes?

Response: The data fusion technique is applied to every pixel regardless of scene type, i.e., all sky.

Page 4, line 10 says: "the scene must be high confi̧dence clear" Is it just the way the validation was done, that is a clear-sky only validation? Same question for the scan angles: "must be less than 50 degrees". Or is it because the data fusion technique only applies to clear-sky, less then 50 degrees pixels?

Response: The comparison between Aqua MODIS and VIIRS fusion radiances was performed only where the cloud detection process for both sensors deemed a pixel to be high confidence clear. This validation, or comparison, approach was also limited to VIIRS scan angles $\leq$ 50ЁŽ, so that fusion radiances were well within the CrIS swath. The text has been updated.

Figure 2 (a) and (b). What do these differences mean? Can the author provide a comparison, on the same fi̧gures, with respect to the instrument noise of the VIIRS and MODIS instrument?

Response: RMS values have been found to be within 1.1 K for the water vapor bands and within 0.5 K for the CO2 bands. Text now indicates this. Text also notes that MODIS radiance comparisons with respect to IASI over six years found that the water vapor bands showed scatter up to 1.0 K in the H2O bands and 0.5 K in the CO2 bands (with a reference to Moeller et al., 2014).

Page 8, line 3: This sentence: "A clear sky regression relationship is established between TPW and VIIRS IR window brightness temperatures (BTs) and the NUCAPS TPW soundings calculated from a global training radiosonde-based profile data set." might not be entirely not clear. What is the training ensemble, what are the predictors?

Response: the predictors are the VIIRS IR window brightness temperatures and the NUCAPS TPW soundings calculated form the training dataset; they are regressed against TPW. The text has been modified accordingly.

Page 8, line 5. Can the author provide more description on the use of the surface emissivity database, when they state: "A high spatial resolution surface emissivity database (Borbas et al, 2018) is used to help differentiate surface emission and atmospheric moisture absorption."

Response: To help differentiate surface emission and atmospheric moisture absorption and to get better surface characteristics in the forward model calculation, surface emissivity for the VIIRS channels used in the regression method has been assigned for each profile in the training dataset from the University of Wisconsin high spatial resolution surface emissivity database (Borbas et al, 2018). The text has been updated.

Page 16, line 10: "CO2 absorption IR spectral bands" is this part of the sentence necessary to the extent of moisture retrieval products?

Response: Both water vapor and CO2 bands are used in the moisture retrievals.

A general comment about Figure 6. The VIIRS+CrIS product improves significantly over the VIIRS only and VIIRS+NUCAPS, in terms of both mean and sdv when compared to the MODIS product. Can the authors explain the impact of this improvement in terms of continuity of the data record. Are there specificities requirements? This remark would strengthen the value of figure 6 and, more importantly, the very final conclusion remark.

Response: The VIIRS+CrIS compares with MODIS TPW within the MODIS product

accuracy of determined from CART site comparison (MODIS Atmospheric Products ATBD, 2011), thus VIIRS+CrIS is a viable source for MODIS moisture product record continuation. Text now states this.

Please also note the supplement to this comment:
https://amt.copernicus.org/preprints/amt-2020-248/amt-2020-248-AC2-supplement.pdf